# Barriers and Facilitators of Implementing a Healthy Lifestyle Intervention at Workplaces in South Africa

**DOI:** 10.3390/ijerph21040389

**Published:** 2024-03-23

**Authors:** Shivneta Singh, Ashika Naicker, Heleen Grobbelaar, Evonne Shanita Singh, Donna Spiegelman, Archana Shrestha

**Affiliations:** 1Department of Food and Nutrition, Faculty of Applied Sciences, Durban University of Technology, 70 Steve Biko Road, Musgrave, Berea 4001, South Africa; ashikan@dut.ac.za (A.N.); heleeng@dut.ac.za (H.G.); evonnes@dut.ac.za (E.S.S.); 2Department of Chronic Disease Epidemiology, Center of Methods for Implementation and Prevention Science, Yale School of Public Health, New Haven, CT 06510, USA; donna.spiegelman@yale.edu (D.S.); archana@kusms.edu.np (A.S.)

**Keywords:** workplace, intervention programmes, facilitators, barriers, implementation

## Abstract

Current evidence indicates that workplace health and wellness programmes provide numerous benefits concerning altering cardiovascular risk factor profiles. Implementing health programmes at workplaces provide an opportunity to engage adults towards positive and sustainable lifestyle choices. The first step in designing lifestyle interventions for the workplace is understanding the barriers and facilitators to implementing interventions in these settings. The barriers and facilitators to implementing lifestyle interventions in the workplace environment was qualitatively explored at two multinational consumer goods companies among seven workplaces in South Africa. Semi-structured in-depth interviews (IDIs) were conducted with ten workplace managers. Five focus group discussions (FGDs) were conducted among workplace employees. The IDI findings revealed that the main facilitators for participation in a lifestyle intervention programme were incentives and rewards, educational tools, workplace support, and engaging lessons. In contrast, the main facilitator of the FGDs was health and longevity. The main barriers from the IDIs included scheduling time for lifestyle interventions within production schedules at manufacturing sites, whereas time limitations, a lack of willpower and self-discipline were the main barriers identified from the FGDs. The findings of this study add to literature on the barriers and facilitators of implementing healthy lifestyle interventions at workplaces and suggest that there is a potential for successfully implementing intervention programmes to improve health outcomes, provided that such efforts are informed and guided through the engagement of workplace stakeholders, an assessment of the physical and food environment, and the availability of workplace resources.

## 1. Introduction

Non-communicable diseases (NCDs) continue to increase globally, with a disproportionate impact on low- to middle-income countries (LMICs) [1]. As the primary cause of death, NCDs account for 41 million deaths annually, representing 74% of all deaths worldwide [2]. Many of these deaths are a result of cardiovascular heart disease (17.9 million deaths per year), cancer (9.3 million), chronic respiratory diseases (4.1 million), and diabetes (2.0 million, including kidney disease deaths caused by diabetes) [3]. On a global scale, it is anticipated that the total number of NCDs will increase by 17% over the next ten years and by 27% in Africa [4]. This emphasises the critical importance of preventative management measures to reduce the escalating impact of NCDs, especially in regions where healthcare resources may be more limited.

In South Africa (SA), a significant health challenge exists in the form of a quadruple burden of disease, which includes HIV/AIDS and tuberculosis, maternal and child mortality, a growing burden of NCDs, and injury-related disorders [5]. This complex health landscape has far-reaching consequences, affecting the quality of life for individuals and imposing substantial healthcare costs both on a personal and national level [6]. In SA, NCDs are the leading causes of death, accounting for 39% of all fatalities in 2010, with more than a third of deaths occurring before age 60 [7]. 

SA is ranked among the top 20 overweight and obese countries worldwide amidst an elevated rate of food insecurity and a high risk of starvation [8]. Obesity affects an estimated 27% of the population, with South African women having one of the highest prevalence rates globally (42%) [9]. South Africa’s obesity prevention and management strategy (2023–2028) was released by the National Department of Health (NDoH) on 3 November 2022, targeting schoolchildren and employees in the workplace with the overarching goal of “creating a healthy food environment where healthier food choices are made easier” [10]. Workplaces are an ideal setting for health promotion [11]. Employees typically spend a substantial amount of time at the workplace; the employed population is relatively stable and is suitable for lengthy health interventions and follow-ups. Additionally, workplaces can provide employees with space and infrastructure to conduct healthy lifestyle programmes; the physical and psychological environment of the workplace has a significant impact on employee health, and work efficiency can be improved with employees’ good health. About 44% of South Africans are employed, and those who work full-time spend more than one-third of their waking hours at work, five days a week [12]. 

According to Pender’s health promotion model, factors that “directly hinder the adoption of health-promoting behaviours or mediate by reducing the commitment to the plan of action for changing behaviour” are considered barriers to a healthy lifestyle [13]. On the other hand, facilitators are factors that promote or enable the uptake and maintenance of a healthy lifestyle [14]. A systematic review conducted by Kelly, Martin [15], identified various barriers to engaging in health behaviours, including insufficient time, lack of accessibility, financial costs, entrenched attitudes and behaviours, restrictions in the physical environment, low socioeconomic status, and lack of knowledge. Conversely, facilitators included the perceived health benefits, such as healthy ageing, enjoyment, social support, clear messaging, and the integration of healthy behaviours into one’s lifestyle. In a qualitative study that examined the facilitators and barriers to implementing a lifestyle intervention in the Netherlands, Tonnon, Proper [16] found that a high level of perceived risk, perceived added value of the intervention, and perceived social support were factors that facilitated employee engagement, while time constraints were an anticipated barrier to implementation. Furthermore, an intervention study to prevent cognitive impairment and disability in Finland highlighted that the facilitators and barriers to implementing a lifestyle intervention programme were associated with infrastructure and resources, personal characteristics, and the nature of the lifestyle intervention [17]. 

Many studies have investigated the effectiveness of lifestyle programmes [18,19,20]. Verweij, Coffeng [21] discovered, for instance, that there were noticeable changes in the reduction of body weight, body mass index (BMI), and body fat. Beyond individual health benefits, employers could benefit from higher productivity rates, reduced absenteeism, and reduced associated costs [22]. Although studies have demonstrated positive results, some lifestyle programmes implemented in the past have not yielded successful results [23]. The challenge of integrating research into real-world settings has been highlighted. Designing workplace interventions without stakeholder buy-in could prevent positive outcomes. For example, inadequate strategies for implementation may result in low employee compliance and participation rates, which would reduce effectiveness [24]. Additionally, a study focusing on identifying barriers and facilitators to implementing a comprehensive workplace health promotion program (WHPP), which addressed various lifestyle factors, indicated that facilitators within the internal environment included accessible resources, knowledge availability, leadership engagement, and consistent communication. Barriers encompassed varied implementation approaches within a single organisation and concerns regarding potential interference with employees’ daily lives [25].

Conversely, in a study conducted on the barriers and facilitators to participation in WHPPs, the primary barriers were identified as an unsupportive organisational culture and programmes that are not contextualised to address employees’ specific needs [26].

This study aimed to identify and comprehend the barriers and facilitators perceived by production and office-based employees at multinational workplaces within fast-paced consumer goods companies in SA. Formative research, including employee input and assessments of employee readiness to implement workplace programmes, is necessary to gain insights for the design and implementation of the intervention that provides an optimal fit into the workplace context. Human resource management plays a vital role in implementing workplace health promotion strategies and managing employee well-being; therefore, this study holds specific relevance for employee management within organisations.

## 2. Materials and Methods

The barriers and facilitators to implementing lifestyle interventions in the workplace environment was qualitatively explored at two multinational consumer goods companies in South Africa. 

### 2.1. Participants

A total of seven consumer goods manufacturing workplaces, with four located in KwaZulu-Natal and three in Gauteng province, were recruited for the study. A total of 2709 workers were employed at the consumer goods manufacturing workplaces that participated in the study. The consumer goods manufacturing company employed 2262 permanent staff, of which 56.5% were male and 43.5% were female. White collar employees dominated at the head office, while each manufacturing workplace had a small proportion of white-collar employees in administrative positions, and the remaining employees were production employees. Another multinational consumer goods and fashion apparel company from the KwaZulu-Natal province was also recruited. The multinational consumer goods and fashion apparel company employed 447 workers, of which 31% were male and 69% were female employees. Among the white-collar employees, 425 were permanent, and 22 were contract employees. 

Semi-structured in-depth interviews (IDIs) were conducted with ten workplace managers from five of the seven workplaces (two workplace managers per workplace). Purposive snowball sampling was used to recruit workplace managers. Participants included full-time staff in a management or supervisory position. A stakeholder map was also used to identify suitable participants for the study. A stakeholder map is a visual depiction of key stakeholders and their associations illustrated on a map [27]. Stakeholders are categorised by factors such as interest, influence, and power and are important for stakeholder analysis. Recognising and considering stakeholders, understanding their perspectives, and actively involving them is widely recognised as a pivotal aspect of any well-structured change initiative [28].

### 2.2. Instruments and Procedure

Five Focus Group Discussions (FGDs) were conducted, each group comprising 4–6 employees, until data saturation was reached to understand the barriers and facilitators of implementing a healthy lifestyle intervention at workplaces in South Africa. The research team created focus group questions, which were reviewed by the researchers for quality and readability. FGDs were guided using a semi-structured FGD guide. To maintain constancy and quality in the data collection, a fieldworkers’ training manual was developed and used to train the research assistants (RAs) before data collection. The manual covered topics such as the nature of IDIs and FGDs, ethical considerations, procedures, and the roles and responsibilities of the team members during data collection. Flyers were distributed strategically at canteen sites, requesting that individuals interested in participating contact the researcher. Participants were purposefully recruited to identify information-rich individuals who represented diverse groups at the workplace (different genders, age groups, and positions). 

The researcher explained the study objectives, risks, and benefits to the participants, as well as the time contributions required from participants. Written informed consent was obtained from all participants. Privacy and confidentiality were assured during and after the FGDs, and the data were de-identified. A digital recorder was used to record the FGD, and a note-taker transcribed the conversation. The recordings were used for quality checks, transcription, and translation and were to be deleted after one year of the completion of the study. 

In-depth interviews and FGDs were conducted online and face-to-face. Six IDIs and three FGDs were conducted face-to-face, while four IDIs and two FGDs were conducted online. The face-to-face interviews were scheduled at a convenient time for the participants, and participants’ contributions were estimated at 45 min for the IDIs and 40 to 60 min for the FGDs. Online IDIs and FGDs were conducted using Microsoft Teams. The scheduling of online data collection activities and consent were obtained via email before the IDI and FGD. The interview was conducted by a trained moderator and supported by a notetaker using a semi-structured interview guide. A digital recorder was used to record all discussions. The recordings were used for quality checks, transcription, and iterative data collection. 

The data consisted of translated verbatim transcripts of the IDIs and FGDs. A thematic analysis technique was employed for the data analysis [29]. This method involved having knowledge, generating initial codes, searching for topics, defining and naming topics, and producing the report. Inductive coding was used to allow results to emerge from the frequent, dominant, or essential themes inherent in the data. The data collection had its own codebook, which was generated, verified for inter-coder reliability, and used to code the transcripts. For example, participants from the IDIs indicated that the best times to conduct health classes at the workplace would depend on an employee’s schedule, their type and mode of work, and their preference for lifestyle classes at the end of the day. Therefore, the resulting theme was organisational situational factors such as availability and peak production periods. Coding enabled the researcher to identify emergent thematic elements that could inform the intervention. The emergent themes were related to our research questions as to how the behavioural intervention strategy was most likely to be implemented successfully: 

IDIs:The organisation of the lifestyle classes (location, time of day, composition of groups).The motivators and barriers to employee participation in the lifestyle classes.Factors to consider when designing a healthy lifestyle programme for the workplace and employees.

FGDs:Organisation of the lifestyle classes (location, time of day, composition of groups).The facilitators and barriers to employee participation in the lifestyle classes.

### 2.3. Ethical Considerations 

This study followed a normal protocol for approval by the DUT Institutional Research Ethics Committee (IREC 078/20). The principal investigator (PI) of the project approached industry (gatekeeper letter) to participate in the study through in-person and online meetings. Employees were briefed about the study and the various phases of the study (information letter). All participants who were willing to participate in the study provided informed consent. Participants were advised that there was no financial gain and that they would also incur no costs for participating in the study. All information collected from the participants was kept confidential, and participants were made to understand that their participation in this study was on a voluntary basis with the right to withdraw at any time. Permission to record the IDIs and FGDs was obtained from participants. Participants were told that the dialogue was anonymous despite being taped and that there was no information in the transcribed notes of the IDIs and FGDs that would allow individual subjects to be connected to particular statements. 

## 3. Results

The data collected from the IDIs and FGDs were tabulated into a narrative summary to understand the barriers and facilitators of implementing a healthy lifestyle intervention at workplaces in South Africa. The emergent themes with corresponding representative quotes are presented in Table 1 and Table 2, and a detailed summary is shown in Appendix A. 

The main facilitators for participation in the programme were incentives and rewards and promoting the benefits (positive outcomes) of the programme. At the same time, a major barrier included scheduling time for lifestyle interventions within production schedules at manufacturing sites.


*“Gifts, rewards, and incentives.”*



*“Depends on the peak production period of different departments.”*


The most suitable times to hold health classes would depend on organisational situational factors, while health classes during peak production periods are problematic. Due to shift work, there is a variance in staff availability, which makes it challenging to plan and implement the health classes. Multiple sessions should be held and repeated to accommodate all personnel and overcome these difficulties. 


*“Different departments and individuals will have different work schedules.”*


Participants suggested that the training room and/or gym area, which is well-ventilated and spacious, would be an ideal location to conduct lifestyle classes. 


*“The training room for the educational component of lifestyle classes and the gym (includes a studio) for physical activity. It is well-ventilated and spacious.”*


Due to the COVID-19 pandemic, there was a sudden migration of employees from the office, which spurred the development and adoption of technology that enabled many employees to work remotely and remain productive. Most participants indicated that a multimodal approach should be used to conduct lifestyle education classes. 


*“Face-to-face and online.”*


There were several barriers to participation at the workplace. These included production deadlines and a fear of maintaining confidentiality. Group dynamics were also identified as a barrier. Ineffective group dynamics can have a negative impact on performance, which can lead to a negative outcome for the collective purpose. Participants highlighted that these barriers should be considered when designing the lifestyle programme.

Healthy longevity and lifestyle class scheduling were highlighted as factors facilitating employee participation in the lifestyle program. The accomplishment of a long existence is longevity; however, living to a ripe old age may be hindered by disability or disease. 


*“It’s going to be good for my health.”*


Participants highlighted that certain factors should be taken into consideration when designing a lifestyle programme, such as multimodal lifestyle classes with proper scheduling from managers, accommodation of staff on leave, the size of the class depending on the size of the training venue, and lesson times being communicated in advance. 


*“Online classes would work for the company because people have less time or are travelling.”*



*“Depends on an individual’s lifestyle. For me, I’d prefer face-to-face.”*


The two main barriers to participating in the lifestyle programme were time limitations due to job responsibilities and self-efficacy. 


*“Time.”*



*“The feeling of being tired and not wanting to go back to do work.”*


## 4. Discussion

According to the Ottawa Charter for Health Promotion [30], enabling individuals to exert more control over and improve their health is known as health promotion. An individual or group must be able to identify and achieve goals, meet needs, and adapt to or cope with the environment to achieve total physical, mental, and social well-being. Given that the workplace represents an ideal setting for the promotion of health, it is crucial that intervention design be firmly rooted in stakeholder input for active participation and successful adoption of the intervention.

One of the main facilitators for participation in the programme was incentives and rewards. An important distinction can be made between extrinsic rewards (such as financial incentives) and intrinsic rewards (pleasure, satisfaction, etcetera) [31]. The reward serves as a motivator for maintaining a new behaviour, making it a crucial element in the repetitive implementation of the behaviour [32]. According to a survey conducted in 2012, 35% of organisations with more than 50 employees offered financial incentives to promote participation in wellness initiatives [33]. Employers intend to encourage behaviours associated with employee well-being while decreasing healthcare costs through these programmes. The ability of these initiatives to promote regular exercise is linked to increased physical functioning and lowered risks for type 2 diabetes and cardiovascular disease. 

The main factor facilitating employee to participation in the lifestyle programme was perceived benefits such as healthy longevity. The processes that govern longevity and life expectancy are influenced by genetic, environmental, behavioural, and dietary variables. The research reported in a review [34] showed that specific diets encouraged the development of healthy longevity by modifying the biological processes connected to ageing and delaying the start of the major NCDs, thereby lengthening lifespan. Employee participation in lifestyle programmes is encouraged as a result of increased employee awareness of the effects of people living with disabilities as a result of NCDs. A particularly meaningful measure of the impact of NCDs is the number of healthy years of life lost due to the diseases. That measure can be calculated in terms of disability-adjusted life years (DALYs), the sum of productive life years lost to premature mortality and disability [35]. 

Organisational situational factors must be considered when designing a lifestyle programme, as they would facilitate participation. Participants highlighted that the most suitable times to hold health classes would depend on availability, while health classes during peak production periods are problematic. Due to shift work, there is a variance in staff availability, which makes it challenging to plan and implement the health classes. Multiple sessions should be held and repeated to accommodate all personnel and overcome these difficulties. Field McHugh et al. [36] state that 12 h shifts and shift work are standard in the manufacturing sector. During peak production, the demand for goods is significantly high [37]. To meet these demands, manufacturing sites must maximise their productivity. Therefore, health classes should not be scheduled during peak production periods. Organisational maturity is a measure of the quality of a company’s operations. A company with a high maturity level can face challenges and seize opportunities. Thus, a company with a high maturity level would most likely be supportive of implementing the lifestyle programme. 

Managers are vital stakeholders when a workplace establishes a wellness programme as they make choices that may affect employee involvement; for example, managers must develop wellness policies and programmes and allocate resources to assist the implementation [38,39]. Managers may foster a culture of wellness in their organisations by publicly supporting employee health and approving wellness initiatives. According to previous studies, employees are more inclined to engage if the workplace culture promotes personal health and well-being [39]. Communication is vital to ensuring good group dynamics, as there is a higher possibility of conflict, stress, and misunderstandings in the workplace [40]. 

Participants suggested that a spacious, well-ventilated room would be the ideal setting for encouraging employees to engage in healthy lifestyle programmes. Given that the healthy lifestyle programme included physical activity, space was required. Kostrzewska [41] highlights that certain spatial conditions and requirements, such as a spacious environment, must be met for physical activities to be safely and comprehensively engaged in. Most participants also indicated that a multimodal approach should be used for lifestyle education classes. The lifestyle classes could be conducted face-to-face or online. Employees prefer web-based healthy lifestyle classes due to flexibility [42]. According to Parker, McArdle [43], providing a variety of alternatives, such as face-to-face group health coaching from nursing staff, has the potential to increase participant involvement and motivation even though online programmes are economical, convenient, and can be administered virtually [44]. 

While numerous factors support participation at the workplace, there are also various barriers to engagement in the workplace. These include production deadlines, technological challenges, and the fear of maintaining confidentiality. Participants also highlighted that these barriers should be considered when designing the lifestyle program. According to Safi and Cole [45], the most frequently reported barriers to workplace physical activity participation include time constraints, a lack of management support, inadequate facilitation, work imbalances, and cultural factors. These barriers must be considered in designing and implementing any workplace wellness intervention for success. 

Group dynamics can affect participation in the lifestyle classes and accomplish the goals established by the group when there is an optimistic dynamic involved [46]. Ineffective group dynamics can have a negative impact on performance, which can lead to a negative outcome for the collective purpose. According to a study by Nackers, Dubyak [47], the results indicated that group dynamics were related to adherence, attendance, and successful weight reduction. Support from managers and colleagues could significantly impact employee participation [48]. It is essential to have engaging lessons, as they lead to positive emotions and motivation to participate in a group. Engaging lessons are important because they allow participants to interact with each other, promote positive experiences, enable active learning, and drive group engagement.

In the FGDS, similar findings were made. The two main barriers to participating in the lifestyle programme were time limitations and self-efficacy. In a study conducted by Person et al. [49], the common barriers to participation in a workplace wellness programme were inadequate incentives, undesirable location, time constraints, lack of interest in the topics discussed, indistinct objectives, schedule, marketing, unfavourable perspectives on health, and lack of enthusiasm for the programme. Self-efficacy is an individual’s confidence in their ability to plan and conduct the necessary actions to achieve desired attainments [50]. Factors that boost self-efficacy include receiving feedback on an individual’s performance and gradual modifications in behaviour. According to the results obtained from executives working across public and private sector organisations in India, it was revealed that individuals with higher levels of self-efficacy and resilience are inclined to demonstrate a positive outlook towards workplace well-being [51].

The study includes several strengths and limitations. All employees across designations were encouraged to participate in formative assessments to create inclusivity and guide the selected interventions. Due to the formative nature of the design to advise the optimal workplace wellness intervention programme, the results obtained may be regarded as being organisation-specific. Consequently, these programmes will require periodic review and updates to the organisation’s strategic plan to maintain relevance.

## 5. Conclusions

The findings of this study suggest that there is potential to implement intervention programmes to improve health outcomes successfully, provided that they are informed and guided through workplace stakeholder input, an assessment of the physical and food environment, and available workplace resources. Workplace stakeholder input in the design of a healthy lifestyle intervention provides valuable insight into the specific needs and challenges perceived by employees and the practicality of pairing these needs with the available workplace resources for the design of optimal workplace intervention programmes. The identified barriers and facilitators assist in contextualising interventions for optimal fit at the workplace. Incorporating these considerations into the intervention design would lead to improved engagement and acceptance, a stronger sense of ownership, enhanced sustainability of the intervention, and the efficacy of health intervention programmes. Targeted strategies can lead to cost and time savings and focused outcome-driven programmes. It is recommended that a rapid analysis tool for stakeholder input be developed to inform the design and implementation of workplace lifestyle intervention programmes. 

## Figures and Tables

**Table 1 ijerph-21-00389-t001:** Summary of barriers and facilitators from individual interviews with managers for the successful implementation of a lifestyle programme.

Facilitator/Barrier	Pre-Defined Domains	Themes	Summary	Representative Quotes
Facilitator	What would be the best time to hold health classes at your workplace?	Organisational situational factors: Availability Peak production periods	Best times will depend on an employee’s schedule, their type and mode of work, and their preference for lifestyle classes at the end of the day.	“Different departments and individuals will have different work schedules.” “It will depend on COVID-19 level restrictions as staff work online from home during certain levels.” “The best times is production driven.” “Preferably at the end of a workday.”
Barrier	What would a be a poor time to hold classes at your workplace?	Organisational situational factors: availability Peak production periods Stakeholder engagement and communication	Lifestyle class scheduling must not infringe on department peak production periods.	“Depends on the peak production period of different departments.” “This should be communicated well in advance, maybe 6 weeks before to the production manager.”
Facilitator	What would be a good location for the health classes at your workplace?	Workplace resources	The training room and gym area are well ventilated and spacious enough to conduct the health classes. Participants also highlighted that the physical classes can be conducted outside the workplace building in an open space.	“The training room for the educational component of lifestyle classes and gym (includes a studio) for physical activity. It is well ventilated and spacious.” “We have quite a lot of space in front (outside the clinic area) to conduct the physical activity.”
Facilitator	How do you think staff would feel about conducting online compared to face-to-face lifestyle education classes?	Role of technology: multimodal	Both online and face-to-face due to the hybrid work model adopted through the advent of COVID-19.	“Face-to-face and online.” “Employees who don’t have access to resources such as the internet prefer face-to-face.” “We have a lot of young people here, so I am sure they would do online as well.” “There is Wi-Fi on site so there would be no connectivity issue whilst on site.”
Barrier	Do you think participation in lifestyle classes would be affected by gender, type of job position?	Group dynamics	Participation would not be affected by gender; however, employees may feel uncomfortable in a group with senior management.	“No problem with gender. Some employees prefer privacy and may not feel comfortable in a group.” “From experience, I don’t think that they would be comfortable if I were their manager, and I were part of the group.”
Facilitator	What would encourage your employees to participate in the health classes, and why?	Incentivisation, educational tools, workplace support, and engaging lessons	Rewarding employees, interesting educational tools, support from management and peers, and fun lessons would encourage employees to participate in the health classes.	“Gifts, rewards, and incentives.” “Create attractive leaflets.” “The main aim is to create awareness to encourage employees to participate.” “Make employees aware of the benefits of participating in the programme.” “Make it fun, have a team leader, engage with staff, know their interests, have competitions, offer support and motivate staff, create health awareness.” “Support from co-workers.”
Barrier	What challenges might your employees face in participating in the health classes?	Organizational situational factors, group dynamics, technological challenges, management support	Production deadlines, fear of maintaining confidentiality, feeling uncomfortable participating in a group, connectivity challenges for online classes, and support from managers.	“Production issues.” “Exposure, the concern of maintaining confidentiality, therefore not wanting to share their personal experiences.” “Some of them will feel embarrassed to exercise in front of other people.” “Online classes might have connectivity issues.” “Support from managers.”

**Table 2 ijerph-21-00389-t002:** Summary of resulting themes and representative quotes from employees participating in FGDs about the successful implementation of a lifestyle programme.

Facilitator/Barrier	Pre-Defined Domains	Themes	Summary	Representative Quotes
Facilitator	What would be the best way to organise lifestyle classes at your workplace (location, time of day, composition of groups)?	Scheduling and mode	Multimodal lifestyle classes with proper scheduling from managers; the best time would be during the day around shift change; staff on leave must be accommodated; the size of the class will depend on training venue size; communicate lesson time in advance.	“Online classes would work for the company because people have less time or are travelling.” “Depends on an individual’s lifestyle. For me, I’d prefer face-to-face.” “It should be during the day.” “Throughout the year, so you take your leave as you want.” “The number of people that we have in a group does not matter, but it would depend on the space that we have.” “Provide notices/communicate in advance.”
Facilitator	What would facilitate your participation in lifestyle classes?	Healthy longevity Scheduling of lifestyle classes		“Healthy living.” “It’s going to be good for my health.” “Multiple classes to accommodate different employee schedules.”
Barrier	What would prevent your participation in lifestyle classes?	Time limitations Lack of willpower and self-discipline.	Time limitations due to job responsibilities and self-efficacy	“Time.” “The feeling of being tired and not wanting to go back to do work.” “Laziness.” “Unachievable goals”

## Data Availability

All data generated or analysed during this study are included in this published article (Appendix A).

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
