# Peer review of "Barriers and Facilitators of Implementing a Healthy Lifestyle Intervention at Workplaces in South Africa"

_ijerph, 2024, doi:10.3390/ijerph21040389_

Round 1
Reviewer 1 Report
Comments and Suggestions for Authors
Author Response
Thank you very much for taking the time to review our manuscript. Your time and feedback are much appreciated.
Please see the attachment.

Reviewer 2 Report
Comments and Suggestions for Authors
1) The authors made an attempt to explore the barriers and facilitators to implementing lifestyle interventions at the work environment in two multinational, multi-site consumer goods companies in South Africa.
However, the purpose of the study was not clearly formulated, either in the abstract or in the introduction.
Lines 104-108 provide guidance on the introduction – introduction should briefly place the study in a broad context and highlight why 104 it is important. It should define the purpose of the work and its significance. The current 105 state of the research field should be carefully reviewed and key publications cited. Please 106 highlight controversial and diverging hypotheses when necessary. Finally, briefly mention the main aim of the work and highlight the principal conclusions. As far as possible, 108 please keep the introduction comprehensible to scientists outside your particular field of 109 research.
2) The introduction needs to be improved:
The problem is not described from a theoretical point of view. What research gap is indicated. There is no indication of what the authors will do. How their article will contribute to better management in the working environment which promotes better habits and lifestyles.
Please elaborate on the literature review. The authors simply did not do an in-depth literature review. In particular, the literature on leadership is lacking.
3) the methodology of the qualitative study is quite well described
4) the final conclusions are well formulated, but it may be possible to formulate some more practical recommendations for managers
Author Response

(The authors gave the same response as above.)

Reviewer 3 Report
Comments and Suggestions for Authors
I find the subject of the article very interesting, as all studies aimed at promoting health or preventing disease are of great importance. Knowing how variables influence and which ones should be taken into consideration when developing programmes aimed at improving the health of the general population is of great interest to the scientific community. Therefore, this article is topical and deals with a subject that is of concern to researchers and health intervention professionals. Of course, it is one thing if the subject of the study is interesting and another if the study is suitable for drawing general conclusions.
The introduction deals with the topics and variables to be addressed in the study, and I believe that the need for the study is very well argued. The same can be said of the results section and the discussion, in particular, the latter provides a good basis for the results obtained with other research on the subject. However, the material and method section is very confusing. I believe that in order to make it easier to read and understand, it would be better to divide it into the classic sections of an article, i.e. participants, instruments and procedure. As for the conclusions and the summary, I think that some ideas are stated that cannot be commented on the results and the study carried out. Although they indicate "potential" with the study carried out, they cannot affirm, as they do not design a health programme based on their results, nor do they implement it in order to be able to verify that these affirmations or hypotheses work.
I consider that the study is very poor, because doing the study in two organisations in a qualitative way, what is actually being done is a study of the needs of these organisations, supposedly to design a programme and implement it. Therefore, this study would be much more complete if the parties for whom the study is supposed to be carried out were provided. This also has an impact on the generalisability of the results obtained, because due to the specificity of each organisation it may happen that the results obtained are only valid for those organisations and not for others. All this detracts from the value of the study, which is why I believe that they should contribute to the design and implementation of a programme based on the results obtained in the study.
In a more detailed way there are some elements that I consider should be changed, such as:
1. In the summary, the statement at the end of the summary should be reconsidered.
2. The last paragraph of the introduction I think it has been left out, it has nothing to do with what is being proposed.
3. In the material and method section:
a. Subdivide this section into the classic sub-sections to make it easier to understand.
b. Specify how these two companies were selected.
c. When talking about the instruments, you should provide more examples of them or include them in an annex.
4. Include a section on the possible limitations of your study.
5. Rephrase the conclusions without stating aspects that cannot be generalised from the results obtained or the study carried out.
Author Response

(The authors gave the same response as above.)

Round 2
Reviewer 1 Report
Comments and Suggestions for Authors
Good job in addressing concerns of reviewers.
Author Response
Thank you very much for taking the time to review our manuscript. Your time and feedback are much appreciated. Kindly find our detailed responses below and the corresponding revisions/corrections highlighted/in track changes in the re-submitted files.

Reviewer 2 Report
Comments and Suggestions for Authors
The authors responded to my comments: purpose of the study is clearly formulated. However, the introduction does not indicate who will need these results - please refer them in particular to employee management (HRM). Also, the research gap is not clearly exposed
References - The literature has been improved
Author Response

(The authors gave the same response as above.)

Reviewer 3 Report
Comments and Suggestions for Authors
After reading the authors' response letter, reviewing the document again with the changes made in response to the reviewers' suggestions, I consider that the article meets the minimum requirements to be published in the journal. I continue to believe that a modification in the presentation of the methodology would facilitate a better understanding by the reader. But if the authors prefer not to modify it, I have no problem with it being presented as they wish.
Author Response

(The authors gave the same response as above.)
